# Cell division drives DNA methylation loss in late-replicating domains in primary human cells

Jamie L. Endicott[1], Paula A. Nolte[1], Hui Shen[1] ✉ & Peter W. Laird ®[1] ✉

DNA methylation undergoes dramatic age-related changes, first described more than four decades ago. Loss of DNA methylation within partially methylated domains (PMDs), late-replicating regions of the genome attached to the nuclear lamina, advances with age in normal tissues, and is further exacerbated in cancer. We present here experimental evidence that this DNA hypomethylation is directly driven by proliferation-associated DNA replication. Within PMDs, loss of DNA methylation at low-density CpGs in A:T-rich immediate context (PMD solo-WCGWs) tracks cumulative population doublings in primary cell culture. Cell cycle deceleration results in a proportional decrease in the rate of DNA hypomethylation. Blocking DNA replication via Mitomycin C treatment halts methylation loss. Loss of methylation continues unabated after *TERT* immortalization until finally reaching a severely hypomethylated equilibrium. Ambient oxygen culture conditions increases the rate of methylation loss compared to low-oxygen conditions, suggesting that some methylation loss may occur during unscheduled, oxidative damage repair-associated DNA synthesis. Finally, we present and validate a model to estimate the relative cumulative replicative histories of human cells, which we call "RepliTali" (Replication Times Accumulated in Lifetime).

Age-associated DNA hypomethylation[1–4] is associated with several intertwined spatio-temporal features. DNA methylation loss occurs primarily within PMDs, which largely coincide with late replication timing domains[5–11], are enriched in higher order chromatin compartment B[12], and tend to be associated with the nuclear lamina[7]. Cancer-associated DNA methylation loss[6,7,13] is accompanied by changes in replication timing and 3D genome organization[14]. Replicative senescence alters 3D genome compartmentalization[15–17]. Replication timing, altered in both cancer and aging-associated diseases including progeria[18–20], is purported to maintain the epigenome[21,22], although this relationship may be bidirectional[23].

Epigenetic 'clocks'—models trained upon large DNA methylation datasets to predict either chronological age[24–26] or features of biological aging[27,28]—have emerged as powerful tools in aging research in recent years, facilitated by the affordability of DNA methylation microarrays and the subsequent availability of increasingly large publicly available datasets. DNA methylation clocks have far outperformed other metrics of biological age, such as telomere length and transcriptional signatures. Although much focus is on the epigenetic age acceleration that is observed with a multitude of diseases[28,29], and the slowing or reversal of epigenetic age[30], recent clock iterations have the intriguing ability to estimate chronological age across mammalian species[31,32], likely detecting conserved features of aging. Although there have been recent attempts to retroactively classify underlying clock mechanisms[33], a major limitation to the interpretation of clock results is the lack of understanding of what drives the methylation behaviors of each clock's CpGs. Whether the age-associated changes in DNA methylation actively contribute to aging, or are merely passenger events, remains largely unknown.

By their nature, chronological methylation clocks are not mitotic clocks. The various tissues within an organism have the same chronological age, but are comprised of cell types with different

[1]Department of Epigenetics, Van Andel Institute, Grand Rapids, MI, USA. ✉e-mail: hui.shen@vai.org; peter.laird@vai.org

proliferation rates and replicative histories[34]. DNA methylation clocks calibrated to organismal age therefore need to be impervious to cell type composition differences. This eliminates DNA methylation changes that directly reflect ongoing or past cell division from most epigenetic clocks trained to chronological age using multiple tissues. The process of cell division requires the passage of chronological time, but the two can be unlinked since time can pass without cell division, such as in post-mitotic cells.

It is important to distinguish between replicative history and proliferation rate. Replicative history refers to the cumulative number of cell divisions within a single cell's lineage. Proliferation rate refers to the number of divisions per unit of time, usually as a current, ongoing measure. In the greater context of biological aging, three of nine 'Hallmarks of Aging' are attributable, in great part, to cumulative cell divisions: telomere attrition, stem cell exhaustion, and cellular senescence[5]. Therefore, replicative history is closely tied to biological age and thus an important feature to measure independent of biological measures of chronological time. In light of this, the term 'clock' is a misnomer for estimates of cumulative cell divisions. A 'counter', 'enumerator' or 'tally' would more accurately capture the nature of cell division. However, the term 'epigenetic mitotic clock' has become cemented into the existing literature for proposed DNA methylation-based measures of cell division[33,35].

We have previously identified a hypomethylation-prone sequence signature, PMD solo-WCGW, representing PMD CpG dinucleotides immediately flanked by an adenine or thymine ('W') and located at least 35 bp away from the nearest CpG ('solo')[36] (Fig. 1b). PMD solo-WCGW hypomethylation appears to correspond to the approximate replicative history of various tissue types and malignancies, and we hypothesized that this could be attributed to incomplete maintenance methylation at each cell division[6]. Subsequent analyses by other groups confirmed that methylation at PMD solo-WCGWs is indeed maintained poorly relative to other sequence contexts[37]. However, there has been little direct experimental evidence to establish a causal or mechanistic link between replicative history and PMD hypomethylation, and this interpretation has been challenged by others in the field[38,39].

Here, we show experimental evidence that hypomethylation within PMDs is driven by cell division. Erosion of PMD methylation at the most hypomethylation-prone sequence context, solo-WCGW, occurs progressively in each cell type studied and continues after immortalization until equilibrium is reached at a very low DNA methylation level. We further characterize the roles of gene expression and replication timing in PMD methylation maintenance. Finally, we present a model, RepliTali, trained on cultured primary cells, to infer replicative history.

## Results

### Context-dependent methylation change in response to cell divisions

We used serial primary human cell cultures to closely track the in vitro replication of cell populations. Primary human cells ($n = 7$, Supplementary Data 1) were obtained from the NIA Aging Cell Culture Repository Apparently Healthy Collection, at the Coriell Institute for Medical Research, and cultured under recommended conditions with multiple parallel subcultures originating from the same initiating cells (Fig. 1a) through replicative senescence, tracking cumulative cell divisions (population doublings, PDs) at each passage (methods). At each passaging, a fraction of cells was retained for DNA methylation analysis using the Infinium MethylationEPIC array (Illumina).

Analysis of DNA methylation revealed divergent behavior between non-CGI CpGs within different contexts: CpGs located in PMDs progressively lost methylation, and CpGs located outside PMD boundaries experienced either a slight gain of methylation if they were located near other CpGs ('social'), or a slight loss of methylation if they were isolated 'solo' CpGs (Fig. 1c). For CpGs within PMDs, the rate of hypomethylation appears influenced by immediate context, again with 'solo' CpGs losing methylation more rapidly than 'social' CpGs, and specifically with solo-WCGWs experiencing the most dramatic methylation loss, which is consistent with previous cross-sectional static characterizations in tissues.

We investigated whether PD-dependent PMD solo-WCGW hypomethylation occurs in different cellular contexts. We observed that across a range of primary human cell types from different developmental stages, the median methylation of PMD solo-WCGWs is tightly anticorrelated with PDs (Fig. 1d). The starting median methylation varies across the primary cells, suggesting that the tissues from which these cells were derived have distinct replicative histories—an observation consistent with the variation in donor age and source tissue. In addition, the rates of global PMD solo-WCGW methylation loss vary between cell types, perhaps reflecting different landscapes of CpG behavior. The pattern of methylation loss at individual PMD solo-WCGWs was reproducible between biological replicates (Fig. 1e).

### PMD solo-WCGW methylation loss is driven by proliferation-associated DNA replication

Elapsed time is linearly correlated with PDs until near-senescence for each primary cell culture with a constant rate of cell division (Supplementary Fig. 1a). As a result, methylation at PMD solo-WCGWs also correlates strongly with time (Supplementary Fig. 1b). Therefore, the serial passage by itself cannot distinguish between time-dependent loss of DNA methylation versus hypomethylation driven by cell division. To determine whether PMD solo-WCGW methylation loss is driven by cell division, or merely ensues with the passage of time, we cultured primary human fibroblasts with media containing decreasing concentrations of fetal bovine serum to impose different proliferation rates. We found that decreased rates of cell division by serum deprivation caused a dose-dependent reduction in DNA methylation loss, consistent with proliferation-associated loss of PMD solo-WCGW methylation (Fig. 1f–h). We have previously hypothesized that PMD solo-WCGW methylation loss is driven by incomplete maintenance methylation. Evidence from other groups has found that the solo-WCGW context is maintained inefficiently, although replication-uncoupled methylation was able to compensate somewhat, at least for a single cell cycle[37]. To test whether methylation loss is indeed driven by proliferation-associated DNA synthesis, we transiently treated several primary cells ($n = 3$) for 3 h either with mitomycin C (MMC), a DNA replication inhibitor that can achieve full permanent cell cycle arrest, or with vehicle control, and maintained the cells for several weeks free of drug (Fig. 1i–k). Two of three primary cells did not lose significant PMD solo-WCGW methylation upon DNA synthesis arrest via MMC (one-sided t-test of logit-transformed beta values: AG11182: $p$-val 0.28, AG11546: $p$-val 0.60, AG16146: $p$-val 1.2e−4). Interestingly, MMC-treated adult fibroblast AG16146 did lose a statistically significant amount of methylation at PMD solo-WCGWs, albeit roughly 5x less than the control condition, indicating that these cells may have somewhat higher tolerance for MMC (Fig. 1i, k). Untreated, freely proliferating cells all experienced significant methylation loss ($p$-val <2.2e−16 for each cell) albeit at different levels (change in fractional methylation from pre-treatment 0.048 AG16146, 0.028 AG11182, 0.03 AG11546), again suggesting that these primary cells may have unequal susceptibility to MMC. Despite this, these experiments clearly show that PMD solo-WCGW methylation is lost as a function of cellular proliferation. Importantly, MMC treatment may have effects beyond the blockade of DNA synthesis. However, our results, plus previous mechanistic studies[37], strongly indicate that progressive methylation loss at PMD solo-WCGWs is caused directly by a failure of maintenance re-methylation.

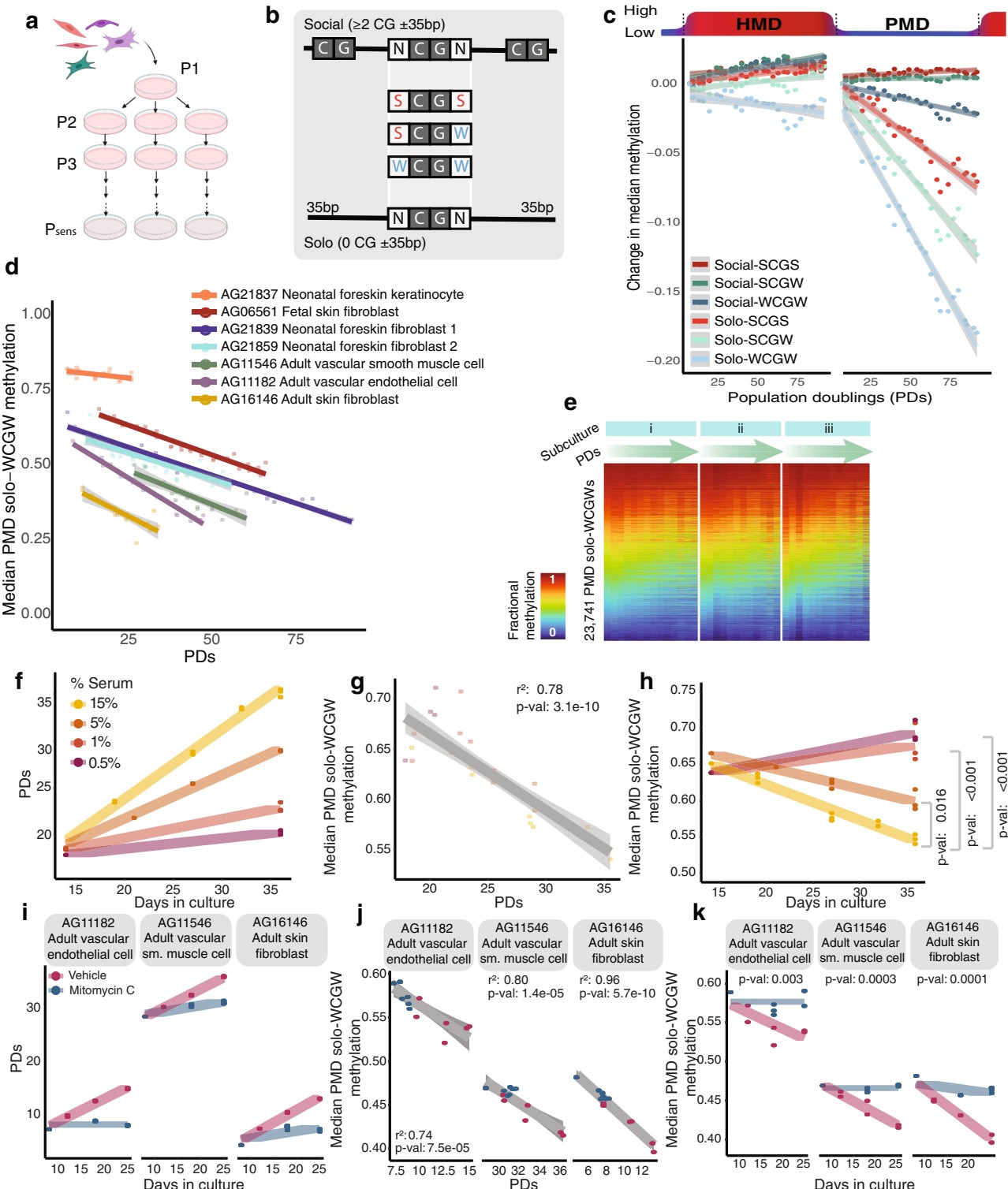

**Fig. 1 | Methylation loss at PMD solo-WCGWs is driven by proliferation-associated DNA replication. a** Schematic illustration of primary cells cultured through replicative senescence. **b** Illustration of immediate (≤35 bp) CpG contexts investigated in this study. **c** Fractional methylation change per population doubling (PD) for neonatal foreskin fibroblast AG21839 at select CpG contexts, within and outside of common partially methylated domain (PMD) boundaries. HMD: highly methylated domains. CpGs within CGIs were excluded. **d** Median fractional methylation of multiple primary cells derived from unique donors and tissues (n = 7) plotted against PDs achieved during this study. **e** DNA methylation heatmap of PMD solo-WCGWs during primary cell culture of neonatal foreskin fibroblast AG21859. Heatmap is separated by parallel subculture, with samples ordered from passage 1 through replicative senescence. **f–h** Primary fetal skin fibroblast (AG06561) grown in media containing different % v/v fetal bovine serum loses PMD solo-WCGW methylation as a function of proliferative rate. **i–k** Primary cells (n = 3) transiently treated with DNA crosslinking agent Mitomycin C for 3 h, followed by drug-free culture for 25 days, resulting in the inhibition of DNA synthesis and subsequent growth arrest (**i**), have stable PMD solo-WCGW methylation. PMD solo-WCGW methylation is tightly correlated to PDs (**j**), independently of time (**k**). Solid lines depict linear regression with gray shading depicting 95% confidence interval; statistical analyses are two-sided. Statistical comparisons for panels **h** and **k** were performed using mixed effects modeling; *p*-values adjusted (Tukey) for multiple comparisons presented in panel **h**.

Taken together, these results present experimental evidence of a direct causal relationship between proliferation-associated DNA synthesis and PMD solo-WCGW hypomethylation.

## Factors driving CpG methylation trajectories in primary and immortalized cells

We investigated factors that could influence the varied rates of methylation loss among CpGs and between primary cell types. Despite the similar profiles of median PMD solo-WCGW methylation loss, we observed subtle cell-type differences at individual CpGs (Supplementary Fig. 2). To explore the possibility that differential expression of maintenance methylation machinery, de novo methyltransferases, or TET enzymes may underpin cell type differences and/or the overall methylation loss, we conducted time-series RNA-seq of our cultured primary cells. PCNA-normalized expression patterns were inconsistent between primary cells and did not clearly accompany the progressive methylation loss we observed in all cultured primary cells (Supplementary Figs. 3, 4).

PMD solo-WCGWs were grouped into major categories (Fig. 2a, Supplementary Fig. 5); those that remained stably methylated through replicative senescence, those that displayed variable methylation (>10% change), and those that were stably unmethylated. Primary cells from chronologically older individuals displayed a smaller stably methylated group, and larger stably unmethylated group (Supplementary Fig. 5). The variably methylated group was the largest for most primary cell types, and was comprised overwhelmingly of CpGs that lost methylation, although a minor subset gained methylation. We further split the variably methylated group for primary fibroblast AG06561 into quartiles of initial methylation levels to visualize the consistency of methylation loss across a spectrum of starting methylation (Fig. 2a, dark right panels).

To test whether there is a meaningful threshold of replicative history at which PMD solo-WCGW methylation stabilizes, primary fibroblasts (AG06561) were immortalized with a lentiviral construct carrying telomerase reverse transcriptase (*TERT*). DNA methylation was profiled at multiple passages following selection for both immortalized and control vector cells.

Immortalized cells achieved drastically higher PDs than did control cells. We terminated the experiment after more than 150 PDs. At the last passage in this experiment, the immortalized cells remained highly proliferative (Supplementary Fig. 6). DNA methylation analysis indeed revealed a threshold at which PMD solo-WCGW methylation stabilized (Fig. 2b), ~40 PDs following replicative senescence of control cells. Although by the end of the experiment most CpGs had dropped to low levels of methylation, a small minority remained stably methylated (Fig. 2c).

We further investigated the distribution of residual methylation in high-PD *TERT*-immortalized cells. We used the methylation state to group CpGs into high, intermediate, and low methylation for early passage, late passage, and late *TERT*-immortalized cells (Fig. 2c). We identified CpGs that were stably methylated or stably unmethylated throughout, and split the remaining variably methylated CpGs into quartiles of terminal methylation values (Fig. 2c). The genomic coordinates of CpGs in each group were analyzed for enrichment of chromatin marks, genomic features, DNA binding proteins, and other characteristics that may explain their behavior (Fig. 2d, Supplementary Data 2). CpGs that were still highly methylated after extended post-immortalization culture had significant overlap with genomic features related to actively transcribed gene bodies. Among the top enriched overlapping features was H3K36me3, which is known to recruit de novo methyltransferase DNMT3B to transcribed gene bodies[40]. CpGs that had achieved low terminal methylation overlapped significantly with features bound by CTCF/cohesin complex members. The loss of methylation at sites bound by CTCF/cohesin complex members in severely hypomethylated immortalized cells is intriguing, given both

the role of CTCF in maintaining chromosomal stability[41,42], and the well-established link between DNA hypomethylation and chromosomal instability in cancer[43–45]. We also observed an enrichment for hypomethylation at sites bound by c-Fos. We have previously shown that the AP-1 binding motif is overrepresented in genomic regions prone to hypomethylation in colorectal cancer[6]. We propose that DNA hypomethylation continues unabated upon *TERT* immortalization until finally reaching a severely hypomethylated equilibrium, in which compensatory de novo methylation offsets further demethylation. We cannot rule out that the observed methylation stabilization in late-culture *TERT*-immortalized cells is caused by selection against cells undergoing further loss of methylation, but we did not observe a slowing of proliferation rate, nor an increase in cell death in immortalized cells with stabilized methylation.

Strong selective pressures are present during cell culture. However, it seems unlikely that such pressures would produce such consistent and reproducible methylation changes at specific sequence contexts throughout the genome, tracking population doublings in multiple cell types. Others have reported that single memory T cells sorted from the same bulk input and clonally expanded into separate colonies all experienced PMD hypomethylation[46].

While we did not find evidence of altered de novo methyltransferase, TET enzyme, or maintenance methylation machinery expression, our analysis cannot rule out the possibility of a mislocalization event of these factors in near-senescence cells leading to methylation loss, as suggested by others[7]. However, our evidence, as well as past static characterizations of PMD solo-WCGWs in vivo[36] and mechanistic findings that methylation at the solo-WCGW sequence context is maintained relatively inefficiently[37], indicates that the overwhelming majority of methylation loss occurs in actively proliferating cells and continues beyond replicative senescence, until an equilibrium is reached at a low stable level of DNA methylation, likely reflecting compensatory de novo methylation offsetting further loss of DNA methylation.

## Replication timing and gene expression

To leverage our high-resolution methylation data into a more complete mechanistic understanding of PMD solo-WCGW dynamics, we regressed methylation to PDs at individual CpGs and compared the rate of methylation change to public replication timing annotations and primary gene expression data.

The short time window for maintenance re-methylation in late-replicating regions is thought to contribute to hypomethylation at PMDs[13]. However, recent mechanistic studies indicate that maintenance methylation continues beyond S phase, uncoupled from the replication fork[37]. Although replication-uncoupled methylation mostly compensates for incomplete replication-coupled methylation following a single cell cycle[37], its efficiency appears strongly influenced by neighboring CpG content, and the cumulative effect over many cell divisions has not been studied. PMD solo-WCGWs located in the regions replicating the latest lost methylation faster compared to those in earlier-replicating regions (Fig. 2e, i). This relationship suggests that PMD methylation loss is indeed driven by poor maintenance methylation, likely because of poor replication-coupled maintenance and subsequent failure of replication-uncoupled methylation. Other features co-occurring with late replication such as chromatin inaccessibility may further explain this relationship.

Enrichment analysis of CpGs that retained methylation at high PDs in *TERT*-immortalized fibroblasts (Fig. 2d) suggested that active transcription protects against replication-associated methylation loss. Although PMDs are relatively gene-poor[47], there are several thousand gene-associated PMD solo-WCGWs on the EPIC array. Methylation change per PD was compared to expression level of associated genes. Indeed, high gene expression was protective against methylation loss (Fig. 2f, j). This relationship was cell-type-specific; genes with

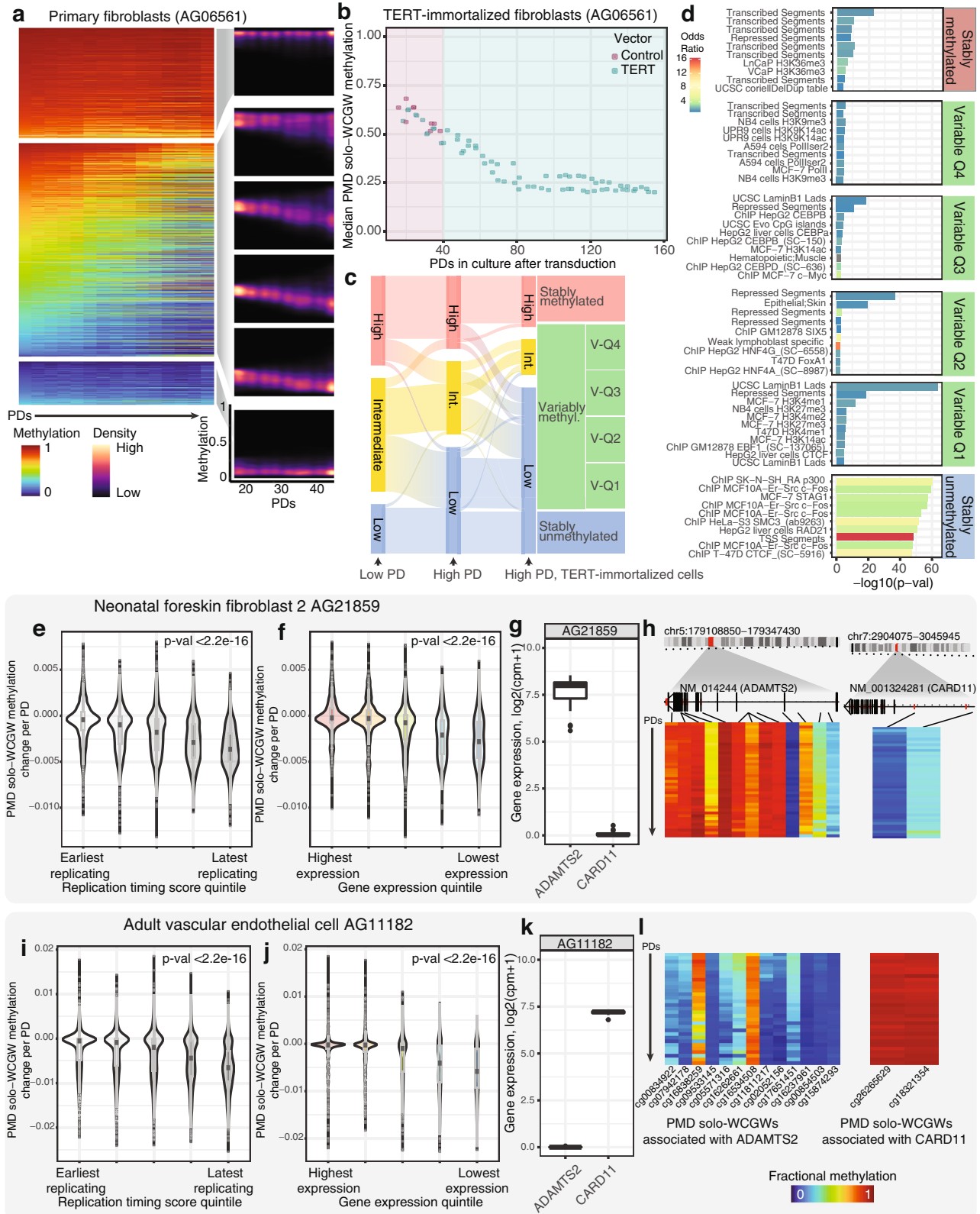

differential expression between fibroblast AG21859 and endothelial cell AG11182 displayed alternate methylation at associated PMD solo-WCGWs (Fig. 2g, h, k, l). Genes with similar expression levels displayed similar methylation (Supplementary Fig. 7). We also examined whether the presence of H3K36me3 influenced the rate of methylation loss (Supplementary Fig. 8). Although there were few array PMD solo-WCGWs overlapping public annotations of this histone mark, its

presence was significantly associated with reduced methylation loss for both cell types.

## Methylation loss during scheduled and unscheduled DNA synthesis

Culture characteristics are arguably non-physiologic; one with particular relevance to longevity research is oxygen exposure[48]. Chronic

**Fig. 2 | Meaningful groupwise PMD solo-WCGW behaviors. a** PMD solo-WCGWs were separated into major categories: (from top) stably hypermethylated, variable, and stably hypomethylated. Representative primary cell AG06561 (fetal skin fibroblast) is depicted. Left, methylation heatmap of CpGs (rows) within each category. Samples (columns) are ordered by advancing population doublings (PDs). Right, density plot of probes within each major category, with the variable group further split into quartiles by starting methylation. **b** Median PMD solo-WCGW methylation for *TERT*-immortalized and control primary fibroblasts through replicative senescence for control fibroblasts (pink-shaded region) and through late PDs for immortalized fibroblasts (blue-shaded region). **c** Redistribution of PMD solo-WCGWs at early, non-immortalized passage, late, non-immortalized passage, and late, *TERT*-immortalized passage. **d** PMD solo-WCGWs in *TERT*-immortalized fibroblasts were grouped by same paradigm in panel **a**. Locus overlap enrichment analysis was performed on each group, with all PMD solo-WCGWs on array as background. **e** PMD solo-WCGW methylation change per population doubling (PD) for neonatal foreskin fibroblast 2 (AG21859) binned into

quintiles based on ENCODE replication timing WA scores from BJ fibroblasts. **f** Methylation change per PD binned into expression quintiles of CpG-associated genes (primary RNA-seq data, AG21859). **g** Fibroblast gene expression for differentially expressed genes *ADAMTS2* and *CARD11*. **h** Fibroblast DNA methylation heatmaps for PMD solo-WCGWs associated with differentially expressed genes *ADAMTS2* (left) and *CARD11* (right). Samples (rows) are arranged from early PD to late PD. **i** PMD solo-WCGW methylation change per PD for adult vascular endothelial cell (AG11182) binned into quintiles based on ENCODE replication timing WA scores from HUVECs. **j** Methylation change per PD binned into expression quintiles of CpG-associated genes (primary RNA-seq data, AG11182). **k** Endothelial cell gene expression for differentially expressed genes *ADAMTS2* and *CARD11*. **l** Endothelial cell DNA methylation heatmaps for PMD solo-WCGWs associated with differentially expressed genes *ADAMTS2* and *CARD11*. Boxplots in panels **e**–**g** and **i**–**k** depict data quartiles; center bar depicts median value. Statistical comparisons for panels **e**, **f**, **I**, **j** by two-sided Kruskal–Wallis test.

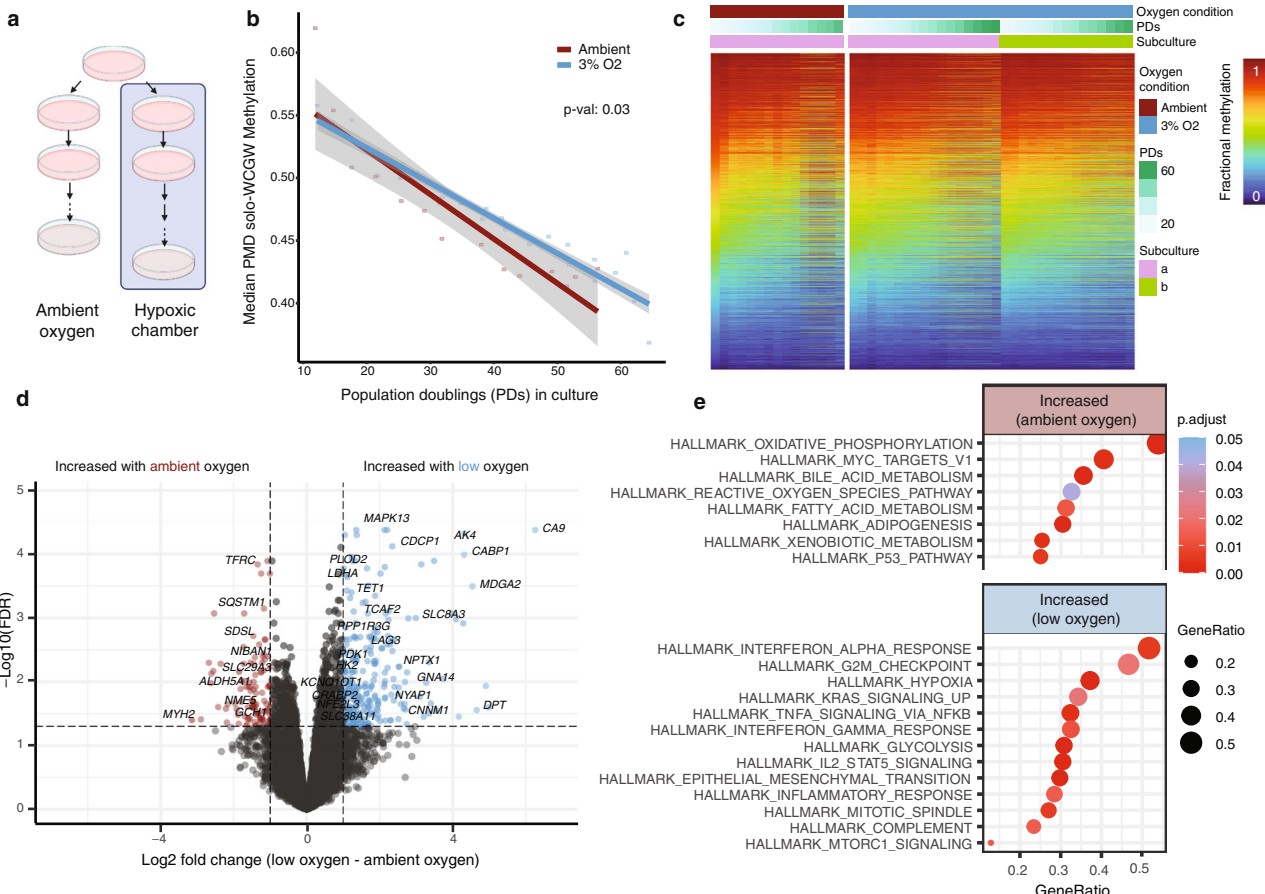

**Fig. 3 | Low culture oxygen slows PMD solo-WCGW methylation loss.**
**a** Schematic of tandem hypoxic/ambient oxygen primary cell culture. **b** Median PMD solo-WCGW methylation plotted against population doublings (PDs) for both culture oxygen conditions. Solid lines depict linear regression with gray shading depicting 95% confidence interval. Statistical comparison of slopes by one-way

ANOVA (*F* = 5.3, two-sided comparison). **c** DNA methylation heatmap of PMD solo-WCGWs for both culture oxygen conditions. **d** Volcano plot of differentially expressed genes between low oxygen and ambient oxygen culture. **e** Pathway enrichment analysis results.

exposure to either high oxygen or reactive oxygen species results in premature aging phenotypes[49,50]. Primary cells grown in hypoxic chambers achieve more PDs[51–53]. To determine whether oxygen partial pressure affects PMD solo-WCGW dynamics in cultured cells, we serially cultured primary fibroblasts (AG21859) under ambient (~20%) and low oxygen (3%) conditions (Fig. 3a), then performed DNA methylation profiling and RNA-sequencing across the series.

Primary cells grown under low oxygen conditions indeed achieved more PDs before replicative senescence than those grown under ambient oxygen conditions (Supplementary Fig. 9).

Interestingly, median PMD solo-WCGW methylation loss was slowed under low oxygen culture (Fig. 3b). Individual CpGs behaved similarly across PDs between conditions (Fig. 3c), suggesting that cells grown in low oxygen conditions simply lose methylation more slowly (Supplementary Fig. 10).

Gene expression analysis between cells grown under both conditions revealed 641 genes significantly upregulated and 373 genes significantly downregulated under low oxygen culture (Fig. 3d, Supplementary Data 3). Top-upregulated genes in the low oxygen condition included many well-known hypoxia markers, such as carbonic

anhydrase 9 (*CA9*) and adenylate kinase 4 (*AK4*), validating the experimental system and accompanying gene expression analysis. Top hits from differential pathway analysis included multiple metabolic pathways, pro-inflammatory pathways activated under low oxygen culture, and reactive oxygen species pathway activated under ambient oxygen culture (Fig. 3e).

While the bulk of DNA synthesis and accompanying DNA methylation maintenance occurs during the cell cycle[54,55], a smaller amount also occurs during unscheduled DNA synthesis (UDS)[56]. UDS-coupled methylation maintenance efficiency is also sensitive to CpGs context[57]. We hypothesize that the accelerated methylation loss at PMD solo-WCGWs cultured in ambient oxygen is caused by incomplete methylation maintenance accompanying UDS as a consequence of increased oxidative damage[58]. This presents a minor caveat to using PMD solo-WCGW methylation as a proxy for replicative history. Conversely, the measure might also be useful to sensitively track cumulative oxidant/DNA damaging agent exposure in slowly proliferating cells or tissues.

### RepliTali: modeling estimates of cumulative cell divisions

While median PMD solo-WCGW methylation correlates strongly with cell divisions in culture through standard replicative lifespans, we developed a more refined metric, which we named 'RepliTali' (for <u>Repli</u>cation <u>T</u>imes <u>A</u>ccumulated in <u>Li</u>fetime) to estimate relative replicative histories of human cells and tissues. PMD solo-WCGWs experience dramatic replication-associated methylation loss and are therefore depleted of methylation with relatively few cell divisions. To access a wider dynamic range of replication-associated methylation loss, we expanded the pool of eligible model CpGs to those in all sequence contexts within common PMDs. Since the total number of cell divisions prior to establishment of primary cell culture in our system is unknown, we envision this tool to be useful as a relative measure as opposed to an absolute benchmark for mitotic history. To adjust for variations in the in vivo replicative histories of the primary cells, we trained RepliTali upon normalized PDs using elastic net regression (Fig. 4a, Supplementary Data 4).

### Comparing RepliTali performance to other models

We applied other published DNA methylation-based 'mitotic clocks'[38,59-61] to our primary cell data (Fig. 4b, Supplementary Fig. 11a). Performance of these models was not as tight and appeared highly cell-type-specific. Interestingly, hypermethylation-based clocks epiTOC2, pcgtAge, and MiAge were vulnerable to cell type differences, whereas epiCMIT[61], a clock that selects the higher estimated mitotic age from either a set of CpGs that gains or a set that loses methylation, performed remarkably well on all cultured cell types. This is particularly interesting as epiCMIT was created exclusively from hematopoietic cell DNA methylation data.

Since the published 'mitotic clocks' were not trained on measured cell divisions, but rather on comparisons between different time-points, it is important to investigate the extent to which RepliTali and these other clocks are reflecting time versus cell division. Our cell-cycle-attenuated and -arrested cell cultures are the best way to disentangle these two factors. Although RepliTali was trained on methylation data from primary cells cultured under standard conditions, it performed very well on growth-attenuated (Fig. 4c) and -arrested (Fig. 4d) primary cells, successfully distinguishing between divisions and time. Other existing 'mitotic clocks' performed inconsistently, with cell type-dependent performance again observed for hypermethylation-based clocks (Fig. 4c, d, Supplementary Fig. 11b, c).

### Validating RepliTali on external datasets

We tested the performance of RepliTali and other clocks on a recent DNA methylation dataset of serially cultured primary fibroblasts[62] (Supplementary Data 5). RepliTali performed strongly across all fibroblasts, correlating strongly with PDs under standard culture conditions (Fig. 4e). It is noteworthy that RepliTali produced a higher estimate for some cells; RepliTali estimates total proliferative history, which for primary cells begins in vivo, long before cell cultures have been established. We also observed differences between the slopes of RepliTali-estimated PDs, suggesting that RepliTali may be best suited to compare relative proliferative histories within a given cellular lineage. As the model was trained upon homogeneous primary cell cultures, it will likely perform best on pure or sorted cell populations, as opposed to the heterogeneous cell composition present in primary tissues. For cells growth-arrested via long-term contact inhibition, RepliTali was very stable. Median PMD solo-WCGW methylation also performed well on this external dataset (Supplementary Fig. 12), supporting its use as a measure of replicative history, perhaps on non-EPIC array methylation datasets. Other mitotic clocks had varied performance, appearing sensitive to variations between fibroblasts (Supplementary Fig. 13). Again, epiCMIT outperformed exclusively hypermethylation-based clocks. Finally, we applied RepliTali and other mitotic clocks to several cell lines that have been extensively profiled (Supplementary Fig. 14). All clocks estimated colon adenocarcinoma-derived cell lines SW480 and HCT15 as having extremely high replicative histories. Curiously, the three hypermethylation-based clocks estimated that IMR90, a cell line initially derived from fetal lung fibroblasts that has been extensively cultured, had a replicative history comparable to low-passage primary skin fibroblast AG06561, whereas RepliTali and epiCMIT estimated higher values.

Whereas RepliTali was calibrated on actual, observed PDs accumulated in culture, other mitotic clocks were created using cancer data[60,61] or normal aging blood[38,59] data, with the assumption that malignant or aged tissues have experienced more cell divisions than non-malignant tissue. However, it is possible that CpGs prone to DNA methylation events co-occurring with, but not directly attributable to increased mitotic history in cancer and aging have been selected into these models. In addition, past mitotic clocks were developed using the Infinium HumanMethylation450 (450 K) array. This may explain why PMD solo-WCGWs have not yet been selected en masse as a tool for estimating mitotic age; they are severely underrepresented on this platform. Approximately 11% of genomic CpGs are PMD solo-WCGWs[36], yet they comprise only 1.5% of 450 K array CpGs. The relatively few ($n = 6214$) on the 450 K array were likely included because they overlap an enhancer or other gene regulatory feature, and thus often do not display the characteristic behavior of PMD solo-WCGWs. PMD solo-WCGWs represent approximately 27% of the probes in epiCMIT's hypomethylation probeset, vastly exceeding the 1.5% represented on the 450 K array. EpiCMIT had arguably stronger performance on our data and on the external dataset than the hypermethylation-based mitotic clocks. By comparison, while PMD solo-WCGWs comprise 18 of 87 CpGs in RepliTali, their mean coefficient weight was −3.35, versus a mean coefficient weight of −0.73 of all RepliTali CpGs, indicating that PMD solo-WCGWs contribute heavily to the model. In addition, RepliTali CpGs recapitulated the progressive methylation loss behavior of PMDs at large (Supplementary Fig. 15).

Epigenetic clocks, representing models based on the methylation status at typically dozens to hundreds of CpGs, have become ubiquitous. Despite the astounding power of these models to predict features associated with biological aging—and its reversal[30]—the biological underpinnings of the CpGs that make these clocks 'tick' are often poorly understood. RepliTali is a DNA methylation-based estimator of replicative history. Among methylation 'clocks' it is unique both in its construction—finely tuned upon serially passaged primary cells—and in our understanding of its driving mechanisms. RepliTali outperforms other models both on our own data and on an extensive external dataset. A challenge of developing a methylation clock to track mitoses is the highly variable rates of cell divisions between tissues[34]. However, the ability to dissect replicative history from other aspects of biological aging (perhaps simultaneously measured by

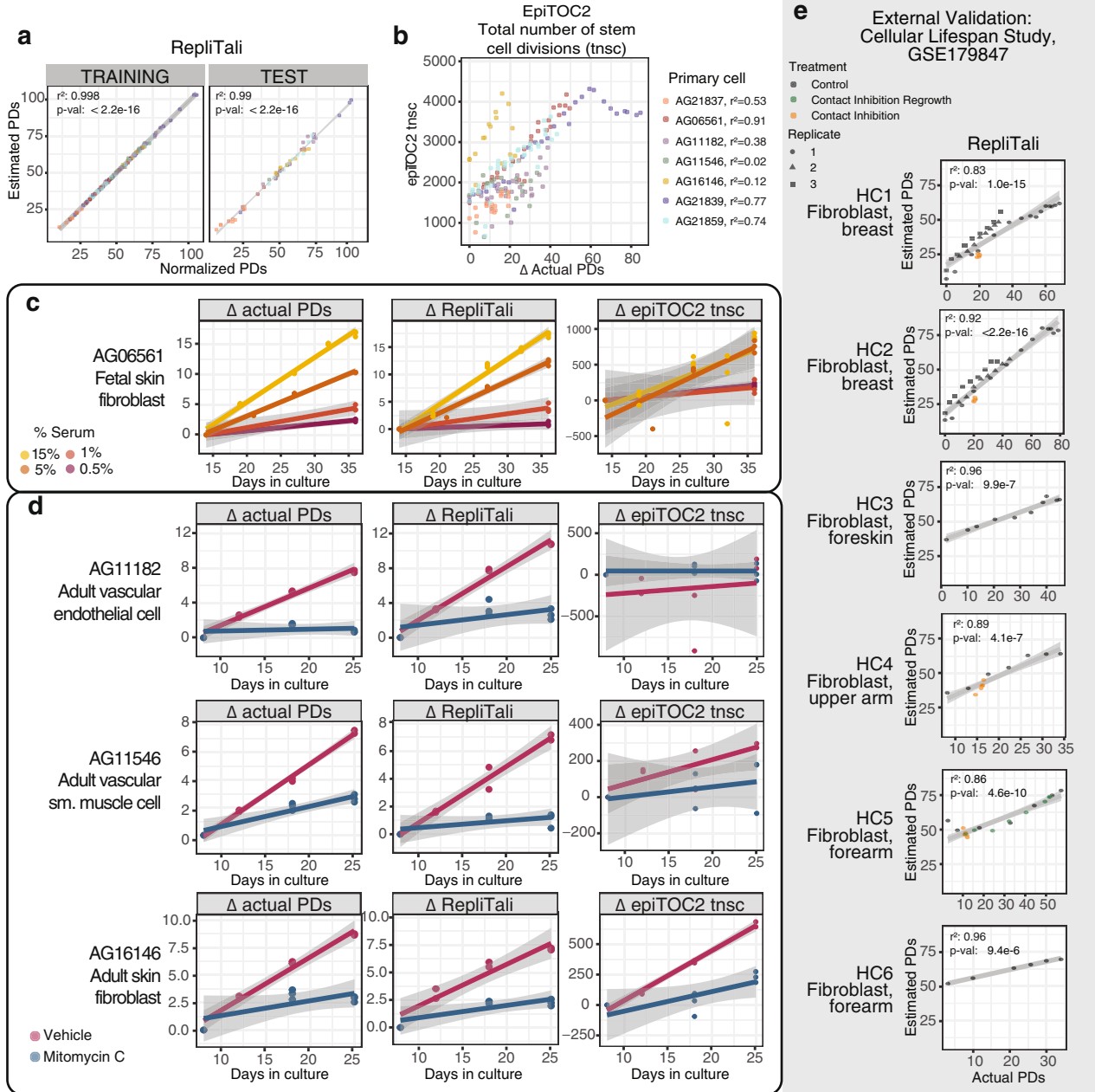

**Fig. 4 | Construction and performance of RepliTali. a** Performance of RepliTali on randomized training (*n* = 122) and test (*n* = 60) sets. Population doublings (PDs) were normalized using a model trained on chronologically youngest cell AG06561 to correct the starting passage PD, with the change in actual PD added to this value for subsequent datapoints. Solid lines depict linear regression with gray shading depicting 95% confidence interval. **b** Performance of epiTOC2, a hypermethylation-based mitotic clock, on primary cell culture DNA methylation data. **c** Model performance on primary fibroblasts (AG06561) grown with different concentrations (%v/v) of media serum to achieve different proliferation rates. **d** Mitomycin C (MMC) treated primary cells (*n* = 3) derived from unique donors and tissues. **e** Performance of RepliTali on external methylation dataset of cultured fibroblasts (*n* = 6 unique donors). Solid lines depict linear regression with gray shading depicting 95% confidence interval; statistical analyses are two-sided.

---

another methylation clock) will aid our understanding of the aging process and inform therapies that seek to slow or reverse it.

## Methods

### Primary cell culture

All primary cells were obtained from the NIA Aging Cell Culture Repository at the Coriell Institute for Medical Research and cultured under recommended conditions. Fetal skin fibroblast AG06561 was maintained in Eagle's MEM with Earle's salts and non-essential amino acids (Gibco 11140-050) with 15% v/v fetal bovine serum. Neonatal foreskin fibroblasts AG21859 and AG21839 were maintained in Ham's F12/DMEM 1:1 media

supplemented with 10% v/v fetal bovine serum. Neonatal foreskin keratinocyte AG21837 was maintained in serum-free human epidermal keratinocyte media (MilliporeSigma SCMK001) on collagen IV-coated dishes (Corning 354453). Adult skin fibroblast AG16146 was maintained in Eagle's MEM with Earle's salts with 10% v/v fetal bovine serum. Vascular endothelial cell AG11182 was maintained in Medium 199 with 1X GlutaMAX (ThermoFisher 35050061), 0.02 mg/ml endothelial cell growth supplement (Corning 354006), 0.05 mg/ml sodium heparin (Alfa Aesar A16198MD) and 15% v/v fetal bovine serum on plates pre-coated with gelatin (MilliporeSigma ES006B). Vascular smooth muscle cell AG11546 was maintained under the same conditions as AG11182 with

the exception of 10% v/v fetal bovine serum. All primary cells were maintained at 37 °C and 5% $CO_2$ with ambient $O_2$ unless otherwise noted. Media was changed at minimum three times per week.

Triplicate cultures derived from the same parent plate or vial obtained from Coriell were maintained in parallel through replicative senescence, which was defined in this study as drastically slowed growth (inability to reach near-confluence at 14 days after previous passage) or viable fraction of cells falling below 60%.

Passaging occurred as cells became ~90% confluent. At each passage, one fraction of cells was pelleted and frozen for future nucleic acid extraction. Another fraction was kept in suspension at room temperature and counted on an automated hemocytometer (BioRad TC20) in duplicate. Viability was determined by trypan blue dye exclusion.

Cumulative cell divisions in culture (population doublings, PDs) were determined using the following equation:

$$PD = 3.32(\log 10(\text{cell yield}) - \log 10(\text{viable cell inoculum}) \\ + X, \text{with } X \text{ being the PD of the inoculum}) \quad (1)$$

### Mitomycin C treatment
Primary cells AG11182, AG11546, and AG16146 were reintroduced into culture from cryopreserved early-passage cells. Duplicate subcultures were derived from the initial recovered plate for treatment and control conditions. Cells were treated with DNA intercalating agent Mitomycin C (MMC, Alfa Aesar J63193MA) reconstituted in DMSO at a final concentration of 10 µg/ml. An equal volume of DMSO was added to control subcultures. Both conditions were incubated for 3 h at 37 °C before media containing MMC or vehicle was removed, cells rinsed with PBS, and basal media replaced. Control cells were passaged normally, and growth-arrested, MMC-treated cells were collected on days 18 and 25.

### Primary cell growth slowing
Primary fibroblast AG06561 was reintroduced into culture from cryopreserved early-passage cells. Four parallel cultures were established and were maintained in media containing 15%, 5%, 1%, and 0.5% v/v fetal bovine serum to encourage different rates of proliferation. At each passaging a fraction of cells was retained for DNA methylation analysis.

### TERT-immortalization
Low-PD primary fibroblasts (AG06561) were transduced with purified lentiviral particles containing expression vectors encoding human Telomerase Reverse Transcriptase (*TERT*) and hygromycin resistance marker (AMSBIO LVP1131-Hygro-PBS), or hygromycin resistance marker alone (control, AMSBIO EF1a-Null-Hygro). Following selection with 250 µg/ml hygromycin B, cells were serially cultured either through replicative senescence (control) or in perpetuity (*TERT*). At the time of analysis, nearly a full year after transduction, the immortalized cells remained highly proliferative.

### Low oxygen cell culture
Low-PD primary fibroblasts (AG21859) were cultured in triplicate in either a standard incubator (Panasonic MCO-19AICUVPA) with ambient $O_2$, or dual-gas $CO_2/N_2$ incubator (PHCbi MCO-170M-PA) at 3% $O_2$, through replicative senescence.

### Methylation analysis by microarray
Frozen cell pellets were thawed and lysed using QIAshredder spin columns (Qiagen 79656). Genomic DNA was extracted from each sample using the AllPrep DNA/RNA Mini Kit (Qiagen 80204), then stored at −80 °C before analysis. DNA was quantified by Qubit fluorimetry (Life Technologies). Approximately 500 ng of genomic DNA was bisulfite converted using the Zymo EZ DNA methylation kit (Zymo

Research D5004) then hybridized overnight on an Infinium MethylationEPIC BeadChip (Illumina), in which the genomic DNA molecules anneal to locus-specific DNA oligomers linked to individual bead types. Raw signal intensities were exported as.idat files, which were processed using the R package SeSAMe[63,64]. Of 386 DNA methylation samples run, 14 failed quality control and were excluded from further analysis, producing a final analytical sample count of 372. All DNA methylation data can be accessed through the Gene Expression Omnibus (GEO) accession GSE197512.

### Statistical analysis
Analysis was performed in R software (version 4.1.1). For comparisons of effect of MMC growth arrest and serum-dependent growth slowing on PMD solo-WCGW methylation, mixed-effects modeling (R package 'lme4') was performed, using logit-transformed (m) methylation values. Multiple comparisons were performed via Tukey contrasts. For comparison of the effect of culture oxygen condition on rate of PMD solo-WCGW methylation models were compared via ANOVA, again using logit-transformed (m) methylation values.

### LOLA
Genomic coordinates (hg19) of PMD solo-WCGW probes of interest were subject to Locus Overlap Enrichment Analysis (LOLA) using R package 'LOLA'[65] and LOLACore (hg19) region set database, available here: https://databio.org/regiondb.

Coordinates of all PMD solo-WCGW probes on the InfiniumEPIC Methylation array were used as background for enrichment analysis.

### RNA-seq
RNA was isolated from frozen cell pellets using the AllPrep DNA/RNA Mini Kit (Qiagen 80204), then stored at −80 °C before analysis. RNA Libraries were prepared from 100 ng of total RNA with the KAPA Stranded mRNA-Seq Kit (Kapa Biosystems KK8401). Indexed libraries were then pooled and 2 × 50 bp, paired-end sequencing was performed on an Illumina NovaSeq 6000 sequencer to a minimum read depth of 30 M reads/library. Demultiplexing was performed using Bcl2fastq (v1.9.0). Differential expression analysis was performed with standard edgeR and DESeq2 workflow. Senescence timepoints were excluded from differential expression and pathway enrichment analysis for oxygen culture condition experiment. Scripts for RNA-seq analytical workflow, including downstream analysis in R, are available here: https://github.com/vari-bbc/rnaseq_workflow.

### RepliTali construction
Starting PD values of primary cells were normalized using an elastic net regression model with alpha parameter = 0.5 (R package 'glmnet') trained on the chronologically youngest primary cell, fetal skin fibroblast AG06561. Samples from all cells were randomized into training (*n* = 122) and test (*n* = 60) sets; normalized PDs were used to construct the final 'RepliTali'. RepliTali is constructed using array CpGs within common PMD boundaries. Coefficients are presented in Supplementary Data 4.

### Mitotic clock comparisons
EpiTOC estimates were obtained using the R script available at: https://zenodo.org/record/2632938#.YdWva5DMKrc. Script was run separately on each primary cell culture, per the author's specifications. Of note, SeSAMe methylation array processing is more stringent than Minfi, hence the suggestion of specifying *p*-val = 0.1 for SeSAMe processing. Care must be taken to evaluate clock CpG dropouts. MiAge estimates were calculated with materials deposited here: http://www.columbia.edu/~sw2206/softwares.htm. epiCMIT estimates were calculated as described in https://duran-ferrerm.github.io/Pan-B-cell-methylome/Estimate.epiCMIT.html.

### Replication timing

Replication timing data from BJ foreskin fibroblasts and HUVECs was generated by the University of Washington and maintained by ENCODE. Files are available here: http://genome.ucsc.edu/cgi-bin/hgFileUi?db=hg19&g=wgEncodeUwRepliSeq. Replication timing weighted average (WA) scores were calculated as previously specified[66]: $WA = (0.917*G1b)+ (0.750*S1) + (0.583*S2) + (0.417*S3) + (0.250*S4) + (0*G2)$.

### H3K36me3

Histone ChIP-seq data from neonatal foreskin fibroblasts was generated by Joseph Costello's lab at UCSF/Roadmap Epigenomics Project. Histone ChIP-seq data from HUVECs was generated by the University of Washington/ENCODE project. Neonatal foreskin fibroblast: ENCSR889OUV|GSM817238. HUVEC: ENCSR000DVM|GSM945233.

### DNA methylation data

Infinium MethylationEPIC array data from serially passaged human fibroblasts was generated by Martin Picard's lab at Colombia University (Cellular Lifespan Study 1.0[62], GSE179847). Raw idats were reprocessed as above.

### Reporting summary

Further information on research design is available in the Nature Research Reporting Summary linked to this article.

## Data availability

The data that support this study are available from the corresponding authors upon reasonable request. The DNA methylation array and RNA-sequencing data generated in this study have been deposited in the Gene Expression Omnibus under SuperSeries accession GSE197545. This accession includes both raw and processed data. External public datasets used in this study are listed here: Replication timing – GSM923444 (BJ Fibroblast), GSM923452 (HUVEC); H3K36me3 – GSM817238 (Neonatal foreskin fibroblast), GSM945233 (HUVEC); DNA Methylation – GSE179847 (Cellular Lifespan Study). InfiniumEPIC Methylation probe manifest[67] is available here: https://zwdzwd.github.io/InfiniumAnnotation. Common PMD coordinates, as well as coordinates and characteristics of PMD solo-WCGWs genome-wide and present on the InfiniumEPIC Methylation array are documented here: https://zwdzwd.github.io/pmd. Source data are provided with this paper.

## Code availability

Custom code used in this study is deposited here: https://zenodo.org/badge/latestdoi/516036288.

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

## Acknowledgements

This work was supported by NIH/NIA grant R01AG066764 awarded to P.W.L. and H.S., and by Van Andel Institute. The authors thank Dr. Wanding Zhou, Dr. Benjamin Berman, and Dr. Andrew Teschendorff for valuable input. Finally, the authors thank the Van Andel Genomics Core for providing DNA methylation array and RNA-seq facilities and services.

## Author contributions

P.W.L., H.S., and J.L.E. conceptualized experiments. Cell culture experiments were performed by P.A.N. and J.L.E.; further analysis was performed by J.L.E. and supervised by P.W.L. and H.S. P.W.L. and H.S. secured funding. P.W.L. and J.L.E. wrote and edited the manuscript with input from H.S. and P.A.N.

## Competing interests

The authors declare the following competing interests. Van Andel Institute has filed a patent application with the US Patent and Technology Office (USPTO #16/977565; PCT # WO2019/167029A1)) entitled "Measuring Replication-Associated DNA Methylation Loss", which describes methods for measuring genomic DNA methylation loss that is linked to cellular replicative history. The inventors on this patent application are: Benjamin P. Berman, Wanding Zhou, Peter W. Laird, Jamie L. Endicott, and Hui Shen. This patent application is currently pending. A license for this patent application has been issued to Trudiagnostic. P.W.L. and H.S. serve on the Scientific Advisory Board of FOXO Technologies. P.A.N. declares no competing interests.
