## [Peer Review File · Nature Communications]

Editorial Note: This manuscript has been previously reviewed at another journal that is not operating a transparent peer review scheme. This document only contains reviewer comments and rebuttal letters for versions considered at *Nature Communications*

REVIEWER COMMENTS

Reviewer #1 (Remarks to the Author):

The authors have some interesting observations. For example, the relationship between gene expression and H3K36me3 and resistance to hypomethylation is interesting, as is the idea that “compensatory de novo methylation offsets further demethylation”. However, this is the 3rd time that I have seen this MS. The authors continue to overstate their conclusions relative to previously published work. There might be a contribution here appropriate for Nat Comms. However, the authors need a more balanced assessment of their work vs the published literature.

The authors state “We present here the first experimental evidence that this DNA hypomethylation (at late-replicating regions of the genome attached to the nuclear lamina) is directly driven by proliferation-associated DNA replication.” This is not true. Cruickshanks et al 2013 proposed that hypomethylation is caused by a failure of DNMT1-mediated maintenance methylation in S-phase of near-senescent cells.

The authors also appear to ignore Cruickshanks et al when they state “Expression of genes encoding de novo and maintenance methylation machinery and TET enzymes was not markedly different between cell lines and across PDs”. Surely, Cruickshanks’ observation of mislocalized DNMT1 in S-phase cells is relevant here?

The authors have extended the observations of Cruickshanks and others with their mitomycin C expts. However, mitomycin C is likely not entirely specific and based on their description “a DNA replication inhibitor that can achieve full permanent cell cycle arrest”, mitomycin C (like most genotoxic stresses on primary human cells) likely induces cell senescence. Thus, mitomycin C does not specifically and only inhibit DNA synthesis, so at best is a blunt tool to conclude that hypomethylation is a consequence of DNA synthesis. These expts lend support to the authors model, to claim it as the “first experimental evidence” is an overstatement.

The authors state “RepliTali performed strongly across all fibroblast lines, closely tracking with PDs under standard culture conditions (Fig. 4e)” I am not convinced. As I see it, the data show that “estimated PD” increases with age, but for several of the cultures, there is a poor correlation between actual PD and estimated PD.

Minor Points

The authors refer to “Age-associated DNA hypomethylation...”. To my knowledge, age-associated hypomethylation of PMDs has not been observed in vivo w/ physiological aging. Hypomethylation of PMDs is a feature of senescent cells in culture and cancer cells. Neither of these is aging. Senescence contributes to aging, but there is near-universal acceptance in the senescence field that senescence in culture should not be equated to physiological in vivo aging.

The primary human cultures are not “lines” – unless they are immortalized.

Reviewer #2 (Remarks to the Author):

This revised manuscript has been significantly improved. The key findings are clearly described and illustrated in the figures, and the manuscript is written in a more balanced way. In addition to providing strong evidence for the role of cell division on DNA methylation loss, this study also clarifies a few vague concepts in the field, which is worth to be published.

I have a few questions that can be addressed with additional bioinformatics analysis and discussions in the text.

Regarding the nature of DNA methylation loss at the PMD, an alternative, yet not conflicting explanation is that it may also reflect the consequence of selection during long-term culture. Maybe an analysis of solo-WCGW sites in the context of neighboring CpG sites may shed light on this question.

For the low oxygen culture experiment in Fig. 3, could the CpG sites be further categorized into groups based on their response to the oxygen conditions? Would there be hints on why those groups respond differentially? How would it relate to the biology of aging?

Regarding the RepliTali model:

1. A more detailed description of the method is needed. Most importantly, which model was used to train RepliTali?
2. The authors make an important point that RepliTali can be used as a measurement of replicative history. However, I think the authors should discuss more about when can this model be applied. Given that this model does not appear to account for differences in methylation loss kinetics, this makes me to consider that it may be not suitable for PDs comparisons between different cell types (as performed in Fig. S12), or cells cultured under different conditions (like the situation in Fig 3b).
3. While may be beyond the scope of the study, it would be interesting to know about the extent to which this model could be applied for the measurement of replicative history in vivo. It is my understanding that the PMD DNA methylation may also be influenced by factors such as cellular composition and the extracellular microenvironment, which in turn affects the accuracy of RepliTali. Therefore, discussion on whether this model could be applied in tissue samples is encouraged.

Some minor points:

1. Page 7, sentences with ref 38-40. Why is DNA methylation loss at CTCF binding sites related to chromosomal instability? Could the authors be more explicit?
2. Does the term "Actual PDs" in Fig. 4 mean normalized PDs? If so, maybe using "normalized PDs" is better.
3. Incorrect order of legend for Fig.1 f-h and i-k.

Reviewer #4 (Remarks to the Author):

The authors have significantly revised the manuscript and expanded their experiments and analyses. I only have a few relatively minor remaining questions.

1. Cells still appear to proliferate actively after mitomycin C treatment, making it similar to the decreased proliferation rate under serum deprivation. Consistently, DNA methylation at PMD solo WCGWs indeed decreases during cell culture. It is hard to fully exclude other possibilities associated with passaging time. The authors should be careful to state that "our tightly controlled experiments provide ... that ... DNA methylation alteration that occurs during aging is not a consequence of the passage of time". Besides, the authors should provide statistical test results for stating "...growth-arrested cells did not lose significant methylation at PMD solo-WCGWs".

2. The authors should again provide statistical results to support the statement that "Expression of genes encoding de novo and maintenance methylation machinery and TET enzymes were not markedly different between cell lines and across PDs". There appears to be more than 2 times the expression difference for several factors (Fig.S3). The authors should at least briefly discuss the potential contributions of DNMT3A/B and TETs in this situation.

Minor comments:

1. The figure legends sometimes are not well aligned to the order of figures (e.g. Fig.1f-k).

Nature Communications manuscript NCOMMS-22-20833-T

Response to Reviewers

We thank the reviewers for their thoughtful and helpful comments. Below we provide a point-by-point response to each of the concerns raised by the reviewers.

Reviewer's comments are in blue italic text.

Our responses are in plain black text.

"Our new added text is in black italics with quotation marks."

Reviewer #1

1. *"The authors continue to overstate their conclusions relative to previously published work. There might be a contribution here appropriate for Nat Comms. However, the authors need a more balanced assessment of their work vs the published literature. The authors state "We present here the first experimental evidence that this DNA hypomethylation (at late-replicating regions of the genome attached to the nuclear lamina) is directly driven by proliferation-associated DNA replication." This is not true. Cruickshanks et al 2013 proposed that hypomethylation is caused by a failure of DNMT1-mediated maintenance methylation in S-phase of near-senescent cells."*

We thank Reviewer #1 for these constructive comments, and agree that appropriate attribution and avoidance of overstating claims are important. We have carefully reviewed all of our comments to make sure that we do not overstate our contribution. Importantly, we have removed the word "first" from all of our claims and statements regarding our results.

The observation of global hypomethylation in aging and cancer has been known for more than four decades. During this very long period, there have been various hypotheses and mechanisms put forward to explain this phenomenon, including a role for incomplete maintenance during replication, active demethylation, and even spontaneous, time-dependent deamination. However, for most of these decades, there has been little direct experimental evidence to support any of the proposed mechanisms. Cruickshanks et al 2013 are not the first to *propose* that hypomethylation is caused by a failure of DNMT1-mediated maintenance methylation. Their experiments focused more on DNA methylation changes associated with senescence, rather than proliferation. They showed that replicative senescent human cells exhibit widespread DNA hypomethylation, preferentially at late-replicating, lamin-associated domains, consistent with other prior reports of such hypomethylation in cancer, including from our own group (Berman et al, 2012). They documented loss of DNA methylation in their senescing cell cultures, but they did not tie this to cell division. They analyzed a small number of time points and did not distinguish between the passage of time and cell division (such as in our serum deprivation and Mitomycin C experiments). Indeed, they reported that methylation loss stabilized at near-senescence in IMR90 cells, and did *not*

continue when near-senescent cells were infected with SV40 to escape replicative senescence. This is likely attributable to the fact that culture of immortalized cells was relatively brief and included few time points. In contrast, we show in our TERT immortalization experiments that methylation loss *does* continue when replicative senescence is bypassed until a stable level is achieved (figure 2b). In our current study the majority of methylation loss occurs as the cells are actively proliferating. We do not observe a precipitous drop in methylation close to senescence. Our interpretation of the Cruickshanks paper is that the focus of that paper and the conclusions were quite distinct from our current manuscript, and that they did not show that DNA hypomethylation is *directly* driven by *proliferation-associated* DNA replication. Nevertheless, we appreciate the importance of avoiding overstatement and hyperbole. Therefore, we have removed mention of being "first" from the manuscript to address this concern.

2. *"The authors also appear to ignore Cruickshanks et al when they state "Expression of genes encoding de novo and maintenance methylation machinery and TET enzymes was not markedly different between cell lines and across PDs". Surely, Cruickshanks' observation of mislocalized DNMT1 in S-phase cells is relevant here?"*

Cruickshanks et al 2013 show that DNMT1 is downregulated and mislocalized in near-senescent cells, whereas we show that hypomethylation occurs consistently throughout culture as a function of cell division, as opposed to just nearing senescence. We also do not observe a downregulation of DNMT1 expression throughout the long proliferative phase of our cultures, during which consistent ongoing DNA hypomethylation occurs. However, we cannot rule out a minor contributing role of DNMT1 mislocalization to hypomethylation occurring near replicative senescence. Therefore, we have added the following text clarifying our results in this greater context.

"While we did not find evidence of altered de novo methyltransferase, TET enzyme, or maintenance methylation machinery expression, our analysis cannot rule out the possibility of a mislocalization event of these factors in near-senescent cells leading to methylation loss, as suggested by others⁸. However, our evidence, as well as past static characterizations of PMD solo-WCGWs in vivo³³ and mechanistic findings that methylation at the solo-WCGW sequence context is maintained relatively inefficiently³⁴, indicates that the overwhelming majority of methylation loss occurs in actively proliferating cells and continues beyond replicative senescence, until a terminal level of hypomethylation is reached."

3. *"The authors have extended the observations of Cruickshanks and others with their mitomycin C expts. However, mitomycin C is likely not entirely specific and based on their description "a DNA replication inhibitor that can achieve full permanent cell cycle arrest", mitomycin C (like most genotoxic stresses on primary human cells) likely induces cell senescence. Thus, mitomycin C does not specifically and only inhibit DNA synthesis, so at best is a blunt tool to conclude that hypomethylation is a consequence of DNA synthesis. These expts lend support to the authors model, to claim it as the "first experimental evidence" is an overstatement."*

We agree that MMC is not a precise tool, and may have additional non-DNA-replication effects on primary cells. However, this experiment provides additional supportive evidence that progressive methylation loss at PMD solo-WCGWs is driven by replication-associated DNA synthesis, together with other lines of evidence, such as the serum deprivation experiments. We have added the following text to our summary of the MMC experiment:

“Importantly, MMC treatment may have effects beyond the blockade of DNA synthesis. However, our results, plus previous mechanistic studies³⁴, strongly indicate that progressive methylation loss at PMD solo-WCGWs is caused directly by a failure of maintenance remethylation.”

4. *“The authors state “RepliTali performed strongly across all fibroblast lines, closely tracking with PDs under standard culture conditions (Fig. 4e)” I am not convinced. As I see it, the data show that “estimated PD” increases with age, but for several of the cultures, there is a poor correlation between actual PD and estimated PD.”*

We are grateful to the reviewer for alerting us to our use of imprecise language. We have clarified this section. Additionally, the reported PD is what was annotated in the GEO dataset, and may not reflect previous culturing. We have revised the problematic text to the following (note: revised text has also changed our erroneous use of cell ‘line’; see comment #6):

“RepliTali performed strongly across all fibroblasts, correlating well with PDs under standard culture conditions (Fig. 4e). Notably, we observed differences between the slopes of RepliTali-estimated PDs, suggesting that RepliTali may be best suited to compare relative proliferative histories within a given cellular lineage.”

5. *“The authors refer to “Age-associated DNA hypomethylation...”. To my knowledge, age-associated hypomethylation of PMDs has not been observed in vivo w/ physiological aging. Hypomethylation of PMDs is a feature of senescent cells in culture and cancer cells. Neither of these is aging. Senescence contributes to aging, but there is near-universal acceptance in the senescence field that senescence in culture should not be equated to physiological in vivo aging.”*

Our 2018 manuscript (reference 33 in the manuscript, reference 2 in the response) presents evidence that hypomethylation is observed in aged tissues, and accompanies both aging and malignancy, with a rate of methylation loss associated with approximate inferred replicative history.

6. *“The primary human cultures are not “lines” – unless they are immortalized.”*

We thank the reviewer for pointing this out. We have corrected every instance of our mistake, including in panel b of figure 4.

Reviewer #2

1. *“Regarding the nature of DNA methylation loss at the PMD, an alternative, yet not conflicting explanation is that it may also reflect the consequence of selection during long-term culture. Maybe an analysis of solo-WCGW sites in the context of neighboring CpG sites may shed light on this question.”*

We thank the reviewer for this alternate explanation. Indeed, long-term cell culture experiments likely experience strong selective pressures on an initially heterogeneous population. While our bulk analysis does indeed preclude investigating this alternative directly, we present several circumstantial lines of evidence that suggest the role of selective population changes is at best minimal in driving PMD methylation loss.

Whereas we observe a steady methylation loss at the PMD solo-WCGW context, a clonal event wherein cells with preexisting hypomethylation overtake cells with a lower proliferative capacity and higher methylation, would presumably exhibit a non-linear rate of methylation change, which we do not observe. Our progressive methylation loss in primary cells from multiple tissues, and the evidence of DNMT1 inefficiency at PMD solo-WCGWs presented by Ming et al³, support our model. Another group has indirectly addressed this question; Salhab et al⁴ sorted several memory T cells from the same bulk input, then expanded them into separate, clonally-expanded colonies. Hypomethylation at PMDs was observed in each colony.

However, we strongly agree that it is an important point of discussion and have added additional clarifying text summarizing the above:

“We cannot rule out that the observed methylation stabilization in late-culture TERT-immortalized cells is caused by selection against cells undergoing further loss of methylation, but we did not observe a slowing of proliferation rate, nor an increase in cell death in immortalized cells with stabilized methylation.”

Strong selective pressures are present during cell culture. However, it seems unlikely that such pressures would produce such consistent and reproducible methylation changes at specific sequence contexts throughout the genome, tracking population doublings in multiple cell types. Others have reported that single memory T cells sorted from the same bulk input and clonally expanded into separate colonies all experienced PMD hypomethylation⁴³.”

2. *“For the low oxygen culture experiment in Fig 3, could the CpG sites be further categorized into groups based on their response to the oxygen conditions? Would there be hints on why those groups respond differentially? How would it relate to the*

biology of aging?”

The reviewer presents an interesting question. We performed additional analysis to determine whether there were different groupwise behaviors of PMD solo-WCGWs between the two culture oxygen conditions. A comparison of methylation change per PD, or slope, between conditions at each CpG revealed a general shift towards greater methylation loss per PD under the normoxic condition, rather than subsets of CpGs behaving differently. We present these additional findings in Supplemental Figure 10.

3. *“Regarding the RepliTali model: A more detailed description of the method is needed. Most importantly, which model was used to train RepliTali?”*

We apologize that the construction of RepliTali was not adequately discussed in the main text. To address this deficiency, we have added the following to our main text describing the modeling of RepliTali:

“...we trained RepliTali upon normalized PDs using elastic net regression.”

Additionally, we have provided a fuller description of the model in Online Methods and have compiled training and test data in our Source_Data document for interested readers.

4. *“The authors make an important point that RepliTali can be used as a measurement of replicative history. However, I think the authors should discuss more about when can this model be applied. Given that this model does not appear to account for differences in methylation loss kinetics, this makes me to consider that it may be not suitable for PDs comparisons between different cell types (as performed in Fig. S12), or cells cultured under different conditions (like the situation in Fig 3b).”*

We concur with the reviewer that RepliTali has limitations and regret that these were not fully discussed. We have included the following text:

“Notably, we observed differences between the slopes of RepliTali-estimated PDs, suggesting that RepliTali may be best suited to compare relative proliferative histories within a given cellular lineage. As the model was trained upon homogeneous primary cell cultures, it will likely perform best on pure or sorted cell populations, as opposed to the heterogeneous cell composition present in primary tissues.”

5. *“While may be beyond the scope of the study, it would be interesting to know about the extent to which this model could be applied for the measurement of replicative history in vivo. It is my understanding that the PMD DNA methylation may also be influenced by factors such as cellular composition and the extracellular microenvironment, which in turn affects the accuracy of RepliTali. Therefore, discussion on whether this model could be applied in tissue samples is encouraged.”*

Please see our response to point #4. We are certainly interested about the possibility of applying RepliTali in vivo, but there are definite caveats and limitations. Application of RepliTali to complex tissues will require methylation-based deconvolution methods, and extends beyond the scope of this manuscript.

6. *“Page 7, sentences with ref 38-40. Why is DNA methylation loss at CTCF binding sites related to chromosomal instability? Could the authors be more explicit?”*

We appreciate that the reviewer notes our undeveloped thought, allowing us to correct this issue. CTCF occupancy has long been linked to DNA methylation, and both occupancy and methylation level of CTCF sites are altered within the context of malignancy. We believe our comment on hypomethylation and chromosomal instability is important, particularly with the broader audience of Nature Communications in mind. We have revised our text and added two relevant sources for interested readers:

“The loss of methylation at sites bound by CTCF/cohesin complex members in severely hypomethylated immortalized cells is intriguing, given both the role of CTCF in maintaining chromosomal stability^{38,39}, and the well-established link between DNA hypomethylation and chromosomal instability in cancer⁴⁰⁻⁴²”

7. *“Does the term “Actual PDs” in Fig. 4 mean normalized PDs? If so, maybe using “normalized PDs” is better.”*

We thank the reviewer for noticing this potentially confusing labelling. Normalized PDs were used only in Figure 4a. We have updated the figure to use the term “Normalized PDs,” and have clarified the text in the accompanying figure legend to reflect this.

8. *“Incorrect order of legend for Fig.1 f-h and i-k.”*

We thank the reviewer for noting our incorrect legend order. We have fixed the ordering to match the figure.

Reviewer #4

1. *“Cells still appear to proliferate actively after mitomycin C treatment, making it similar to the decreased proliferation rate under serum deprivation. Consistently, DNA methylation at PMD solo WCGWs indeed decreases during cell culture. It is hard to fully exclude other possibilities associated with passaging time. The authors should be careful to state that “our tightly controlled experiments provide ... that ... DNA methylation alteration that occurs during aging is not a consequence of the passage of time”. Besides, the authors should provide statistical test results for stating “...growth-arrested cells did not lose significant methylation at PMD solo-WCGWs”.”*

We appreciate the reviewer's attention to this matter, and have performed additional statistical testing on our data. Indeed, it appears that the primary cells had unequal responses to MMC, which is important to address. We have added the following text in reference to our MMC experiment to summarize these findings and better characterize the MMC experiment:

“Two of three primary cells did not lose significant PMD solo-WCGW methylation upon DNA synthesis arrest via MMC (one-sided t-test of logit-transformed beta values: AG11182: p-val 0.28, AG11546: p-val 0.60, AG16146: p-val 1.2e-4). Interestingly, MMC-treated adult fibroblast AG16146 did lose a statistically significant amount of methylation at PMD solo-WCGWs, albeit roughly 5x less than the control condition, indicating that these cells may have somewhat higher tolerance for MMC (Fig. 1i,k). Untreated, freely-proliferating cells all experienced significant methylation loss (p-val <2.2e-16 for each cell) albeit at different levels (change in fractional methylation from pre-treatment 0.048 AG16146, 0.028 AG11182, 0.03 AG11546), again suggesting that these primary cells may have unequal susceptibility to MMC. Despite this, these experiments clearly show that PMD solo-WCGW methylation is lost as a function of cellular proliferation. Importantly, MMC treatment may have effects beyond the blockade of DNA synthesis. However, our results, plus previous mechanistic studies³, strongly indicate that progressive methylation loss at PMD solo-WCGWs is caused directly by a failure of maintenance remethylation.”

2. *“The authors should again provide statistical results to support the statement that “Expression of genes encoding de novo and maintenance methylation machinery and TET enzymes were not markedly different between cell lines and across PDs”. There appears to be more than 2 times the expression difference for several factors (Fig.S3). The authors should at least briefly discuss the potential contributions of DNMT3A/B and TETs in this situation.”*

We have performed additional analysis, presented in Supplementary Figure 4, as clarified and expanded in the following text:

“To explore the possibility that differential expression of maintenance methylation machinery, de novo methyltransferases, or TET enzymes may underpin cell type differences and/or the overall methylation loss, we conducted time-series RNA-seq of our cultured primary cells. PCNA-normalized expression patterns were inconsistent between primary cells (Suppl. Fig. 4) and did not clearly accompany the progressive methylation loss we observed in all cultured primary cells (Supplementary Figs. 3,4).”

3. *“The figure legends sometimes are not well aligned to the order of figures (e.g. Fig.1f-k).”*

Please see Reviewer #2, point #8.

Additional Sources Cited in Response

1. Cruickshanks, H. A. *et al.* Senescent cells harbour features of the cancer epigenome. *Nat. Cell Biol.* **15**, 1495–1506 (2013).
2. Zhou, W. *et al.* DNA methylation loss in late-replicating domains is linked to mitotic cell division. *Nat. Genet.* **50**, 591–602 (2018).
3. Ming, X. *et al.* Kinetics and mechanisms of mitotic inheritance of DNA methylation and their roles in aging-associated methylome deterioration. *Cell Res.* **30**, 980–996 (2020).
4. Salhab, A. *et al.* A comprehensive analysis of 195 DNA methylomes reveals shared and cell-specific features of partially methylated domains. *Genome Biol.* **19**, 150 (2018).

REVIEWERS' COMMENTS

Reviewer #1 (Remarks to the Author):

Most of my comments have been adequately addressed. However -

In response to my comment 4, the authors state "RepliTali performed strongly across all fibroblasts, correlating well with PDs under standard culture conditions (Fig. 4e)." Infact, RepliTali and Estimated PDs do not correlate well. For HC3, HC4 and HC5, RepliTali = 20 corresponds to Estimnated PD = 50.

In response to my comment 5, the authors cite their own 2018 MS ref 33. In the abstact of the current MS, they state "Loss of DNA methylation at late-replicating regions of the genome attached to the nuclear lamina advances with age in normal tissues, and is further exacerbated in cancer. As far as I can tell, refs 5-7 do not show data on aging, - only cancer. In my opinion, the idea that aging is associated with global genome hypomethylation is an old dogma based on old technology that has not stood up to more recent sophisticated analysis from the labs of Derrick Rossi, Peter Adams, Margaret Goodell and others. Aside from their own ref 33, I don't think this MS cites a paper that shows a clear genome wide hypomethylation based on whole genome bisulfite sequencing. The text should be moderated to reflect this - or cite several primary refs that show this for physiological aging (not in vitro senescence).

Reviewer #2 (Remarks to the Author):

My comments have been addressed adequately and I support publishing this study. Just one minor complain. I am reviewer 3, but I could not easily find responses to my comments in the rebuttal letter, until I realized that my points were marked as point 3-8 of reviewer 2.

Reviewer #4 (Remarks to the Author):

All my concerns have been addressed. I have no further comments.

Nature Communications manuscript NCOMMS-22-20833-T

Response to Referee 1

We thank Referee 1 for his/her/their careful review of our manuscript. Below we provide a response to each of the two remaining concerns raised by the Referee 1.

Referee's comments are in blue italic text.

Our responses are in plain black text.

"Our new added text is in black italics with quotation marks."

Most of my comments have been adequately addressed. However -

In response to my comment 4, the authors state "RepliTali performed strongly across all fibroblasts, correlating well with PDs under standard culture conditions (Fig. 4e)." Infact, RepliTali and Estimated PDs do not correlate well. For HC3, HC4 and HC5, RepliTali = 20 corresponds to Estimnated PD = 50.

We thank Referee 1 for pointing out the need for further clarification. RepliTali estimates total replicative history, which for primary cells begins *in vivo* in embryonic development within the respective tissue of origin and will be affected by donor age and anatomical site (e.g. sun exposed vs protected). 'Actual PDs' refers to the measured *in vitro* replicative history, which is a measure relative to the culture starting point. The primary fibroblasts from GSE179847 were established from donors of varying ages and anatomical locations. Additionally, fibroblasts HC3, HC4, HC5, and HC6 were already at passage 3-4 after resection (table 1 in Reference 62 in the main text) with an unspecified PD, prior to culturing in this study. The elevated RepliTali estimates reflect all of these factors. We have added the following clarifying text:

*"It is noteworthy that RepliTali produced a higher estimate for some cells; RepliTali estimates total proliferative history, which for primary cells begins *in vivo*, long before cell cultures have been established."*

*In response to my comment 5, the authors cite their own 2018 MS ref 33. In the abstack of the current MS, they state "Loss of DNA methylation at late-replicating regions of the genome attached to the nuclear lamina advances with age in normal tissues, and is further exacerbated in cancer. As far as I can tell, refs 5-7 do not show data on aging, - only cancer. In my opinion, the idea that aging is associated with global genome hypomethylation is an old dogma based on old technology that has not stood up to more recent sophisticated analysis from the labs of Derrick Rossi, Peter Adams, Margaret Goodell and others. Aside from their own ref 33, I don't think this MS cites a paper that shows a clear genome wide hypomethylation based on whole genome bisulfite sequencing. The text should be moderated to reflect this - or cite several primary refs that show this for physiological aging (not *in vitro* senescence).*

The association between increasing age and global loss of DNA methylation in humans and other mammals is not dogma, but an observation that has been documented in dozens of very consistent primary data papers spanning four decades, using both old and new technologies, ranging from LC-MS, HPLC, repetitive element hybridization in Southern blots and PCR-based techniques, Illumina Infinium arrays, and importantly, whole genome bisulfite sequencing. To address Referee 1's concern, we have added citations to strengthen the sourcing of this statement, but these references represent a small fraction of the large body of literature in support of this statement. Beyond the firming up of the referencing, the experimental demonstration in our current paper that cell division leads to loss of DNA methylation is consistent with a proposed loss of DNA methylation with increasing age. I am not sure how we could reconcile replication-associated hypomethylation with the lack of an associated loss of methylation with increasing age.

In our own analysis (which is now Reference 36 in the revised main text) we show in Figure 6a a multiscale analysis of whole genome bisulfite sequence of more than 30 Mb of human Chromosome 16 in a comparison between cell-sorted normal CD4+ T cells from a newborn and that of a 103-year-old individual. The degree of hypomethylation in the aged normal T-cells is striking. This is not a comparison of cancer. In Figure 6b we show loss of methylation in a data set of unsorted newborn PBMCs versus nonagenarians, and a comparison of liver DNA methylation in fetal tissue versus adult, both again showing significant hypomethylation. Figure 6c shows that this hypomethylation starts during fetal development. Figure 6d shows age-associated hypomethylation in skin samples, with an accelerated hypomethylation in sun-exposed epidermis. Figure 6e and 6f show that the lymphoid lineage loses methylation faster as a function of age than the myeloid lineage. All of these analyses are in ***non-malignant***, normal cell types.

We added references to three more recent reports from other groups of age-associated DNA methylation loss in non-malignant tissues, all based on WGBS data (References 10-12 in the revised main text): Briefly, an effort led by Joseph Ecker's group found low PMD methylation in healthy placenta and pancreas in WGBS data (Reference 10 in the revised main text); WGBS profiling of B-cell lineage cells detected extensive heterochromatic methylation loss in memory B lymphocytes, but not naïve lymphocytes and progenitors (Reference 12 in the revised main text); Andrew Feinberg's group observed an age- and sun exposure-dependent methylation loss in nonmalignant skin samples (Reference 11 in the revised main text).